# Multilingualism as a Mirror of Strangeness in the Translation of Contemporary Literary Texts

Cristina Valdés

Department of English, French and German Studies, Universidad de Oviedo, 33011 Oviedo, Spain; cris@uniovi.es

**Abstract:** This paper focuses on the issue of multilingualism in contemporary literary texts, which contain examples of code-switching or words and expressions in different languages, which contribute to placing emphasis on the foreignness and strangeness of the characters or narrators of the stories. This study stems from the edition of a compilation of short narrative and dramatic texts translated into Spanish by authors who build up stories from a position of in-betweenness, rejection, or displacement. In this context, the presence of different languages contributes to revealing the multilingual and multicultural reality that provides the background for the different stories. They are all concerned about manifesting their vital experiences of (un)belonging to a certain labelled culture or identifiable group, often from a diasporic point of view. Some real examples of translation processes will be provided to show the strategies employed to preserve an effect of strangeness on readers, to reveal feelings of (un)belonging, to manifest a variety of identities, or to make explicit culturally marked terms. Translation is then approached from the perspectives of cosmopolitism, diversity, and postcolonial studies, which rely on multilingualism as a signal of a diversified and multicultural identity.

**Keywords:** multilingualism; strangeness; translation strategies

## 1. Introduction

Multilingual literature has experienced significant growth in the last decades as a form of expressing the diversity and intercultural encounters that have commonly taken place in modern societies, particularly among different social groups that live together in the same territory. Contemporary writers of this kind of literature have been, on the one hand, products of these societies and, on the other, show an interest in revealing the tensions that difference produces. Thus, I aim to explore the effects of multilingualism on the translation of literary texts that share the intention to reflect foreignness and strangeness. Questions such as whether multilingualism disappears in the target texts, its effect on reception, and how it influences the final target text will be addressed.

This paper results from a research project[1] conducted for four years (2018–2021) and funded by the Ministry of Science of Spain on the topic of cosmopolitanism and the figure of the stranger. One of the outcomes of the project was the edition of a volume with a compilation of English and French literary texts that were translated into Spanish, which resulted in the publication of the anthology *Extrañezas cosmopolitas. Antología literaria* (Valdés and Prada 2022). The present study was conceived from a dual perspective, considering that "translation and multilingualism are inextricably connected" (Meylaerts 2010, p. 227), so that it examines multilingualism from the point of view of literature and from the point of view of translation, with the intention to explore how the strangeness in the source texts is presented to the target audience, which has become the novelty and main strength of the anthology. This bears some manifest implications for translation as to how to deal with the presence of several languages during the translation process, which highly depends on the features of each individual text and on the author's intentionality and expected effect on the readership. Therefore, the translator makes a thorough analysis of the source text, paying attention to the text type of the given texts, the kind of language or language

variation(s), or the various translation challenges; all these factors are looked at from the perspective of how language contributes to preserving the strangeness conceived by the author of the original literary text.

Focusing on the examination of some literary texts that raise some interesting questions about how to deal with multilingualism in translation, I will choose O'Sullivan's statement (O'Sullivan 2011, p. 176) that "[m]ultilingualism becomes both a product of translation and a problem for translation" as a starting point for reflection, revolving around the idea that the problem may become an opportunity since translating is a challenge for translators and it may fulfil an aesthetic, pragmatic, and ideological function. In Foran's terms, "the translator would therefore seem to act as an agent of sur-vival" (Foran 2016, p. 144). Moreover, in this paper, we approach translation, both theoretically and as a practise process, from the angle of multilingualism in literary production, placing emphasis on the role played by translation in a context of multicultural identities. Recently, regarding this role of translation, Vidal Claramonte (2021b) has claimed that in a translingual world, concepts such as citizenship, belonging, global justice, hospitality, policy regarding difference, governance, or cosmopolitanism are transforming the way we understand life, and, within this context, language is never neutral or innocent. Similarly, Kellman (2017) speaks about 'the translingual situation', which he defines as "the situation of those writers who write in more than one language or in a language other than their primary one".

Bringing both theoretical statements to a common ground, I would like to highlight, firstly, the idea that the choice of a language is never neutral, that is, one or more than one language is selected with an intended purpose in mind, and, secondly, that translingual writers produce multilingual texts, making it more than patent that multilingualism facilitates the expression of difference while it visibilizes the presence of a combination of languages.

Other questions of interest are what we mean by multilingualism in literature and the implications for translators. Particular attention is drawn to what is to be analysed—the role of the writer or the impact on the reader. In a stimulating paper, "Literary Multilingualism", Bar-Itzhak (2019) brings Yildiz's work (Yildiz 2012) to expand on the idea that multilingualism is not new (see also Meylaerts 2006; Corrius and Zabalbeascoa 2011 on audiovisual texts; Mateo 2019 on stage and film musicals; Mateo 2020 on opera libretti). In fact, monolingual practise is "closely tied to the idea of modernity" (Yildiz 2012) and derives from the notion of the nation-state, so that there is a tendency to establish a correlation between one nation-state, a territorial and political entity, and a language, often a major language. In her paper, Bar-Itzhak (2019) refers to Yildiz's notion of 'the postmonolingual condition', a term "to convey the tensions between the still dominant monolingual paradigm and reemerging multilingual practices" (Bar-Itzhak 2019, p. 1). She also points out that some writers "negotiate the paradigm and write in more than one language, transitioning from writing in one language to another, self-translating, or creating literary works that are themselves, to varying degrees, multilingual texts." This practice is applicable to the selection of texts that have been analysed for the purpose of this paper since most writers of the chosen literary texts speak and write in more than one language, either because they have lived in bilingual or multilingual contexts, they belong to migrant communities, or they aim at representing their dual or multiple cultural backgrounds (see also Vidal Claramonte 2022).

Bar-Itzhak (2019) also proposes a frame for the analysis of literary multilingual texts, which helps us obtain a more accurate and precise examination of the texts. She claims that every study of literary multilingualism raises three central sets of questions: on the representational level, on the level of form, and on the level of interpretation, which correspond to the political, aesthetic, and creative dimensions of texts, according to Bar-Itzhak (2019). This distinction of three levels entails different questions to be answered through the descriptive analysis: how each literary text reflects the reality of a multilingual community or society; what are the interlingual relations within each text; what are the political implications of translation; or how can multilingual fiction function as a creative force within each literary text.

Moreover, Delabastita and Grutman (2005, p. 14) have also explored the role of fictional representations of multilingualism and translation, and they reach the following conclusion:

> [F]ictional representations of multilingualism on the one hand, and of translation on the other, ultimately lead us back to a common reality, that is, if we understand 'translation' not just as an abstract or 'technical' operation between words and sentences, but as cultural events occurring, or significantly not occurring, between people and societies in the real world.

That reference to events 'significantly not occurring' points out what Apter (2013) calls "the Untranslatable", fostering "an approach to literary comparatism that recognizes the importance of non-translation, mistranslation, incomparability and untranslatability" (Apter 2013, p. 3). Following Apter, Bertacco (2014) focuses on the translation of postcolonial literatures in multilingual contexts and states that "when foreign words or unfamiliar varieties of English are used and mixed together in postcolonial texts, they are the graphs of the presence of 'the Untranslatable'" (Bertacco 2014, p. 5), a "compositional heterogeneity that disrupts the fictional continuum" (Apter 2013, p. 17).

Therefore, translation results in real cultural events that take place between people and groups and which facilitate the interaction and knowledge between them, going beyond the duality of the source language (L1) and the target language (L2), allowing to visibilize a third language (L3), which "is neither L1 in the ST nor L2 in the TT; it is any other language(s) found in either text." According to Corrius and Zabalbeascoa (2011, p. 117), "we may refer to a third language regardless of whether the language used in the source text, the L3ST, is some kind of neutral or standard, or otherwise stresses the features of intralingual variation." This bears implications for translation theory, as Voellmer and Zabalbeascoa (2014, p. 237) point out:

> Moving beyond the longstanding view that translation involves two languages: L1, the language translated from, and L2, the language translated into, generally regarded as interlingual translation or translation proper (Jakobson 1959), is a first step for translation theory to start accounting for heterolingual texts.

Acknowledging heterolingualism is a significant step to building awareness that the combination of languages is also meaningful in a text, given that the status of the different coexisting languages is rarely (never) the same, and where there is proof of unequal relations between languages, the notion of untranslatability arises (Bertacco 2014, p. 4).

When dealing with literary multilingualism, another key aspect is the role of the writer of multilingual texts and how the type of authorship influences translation (Moreno Tovar 2020). In the case of literature, translating a multilingual text as a heterolingual text, that is, as a text maintaining the foreign words of the original, is a form of bringing the author's voice and his or her fiction together. The strange foreign words, according to Kaminsky (2013), serve to wake readers to the actual meaning of words and to experience being other:

> In my private library, this is one of the great translations of the twentieth century. But the word "translation," to my mind, is misleading.

> This translation (or any great translation, for that matter) is not a mirror. While one appreciates Felstiner's haunting use of German words interspersed with English, this striking and powerful juxtaposition of languages does not happen in Celan's poem. Translation, however faithful, is fiction. So why is Felstiner's use of German a good decision? Because Felstiner's version is only made more striking as we wake to the actual tragic meaning of the strange foreign words—it gives English readers the experience of being other, a voice alienated from language. To realise this is to see clearly that a successful translation, even a very "faithful" one, has no need to mimic the original. It is the poet's process, writes Eavan Boland, that needs to be translated.

When approaching the nature and type of text in the selection of literary texts that are studied in this paper, an element that merits attention is the writer, who can often be considered a "translingual writer", defined by Vidal Claramonte (2021b, p. 14) as "creators

from any part of the world who do not express themselves necessarily in the language of their birthplace, as writers of multiple belongings, of liquid identities, from Creole archipelagos, nomad and deterritorialised writers, who remind us that one language is many languages, that each of us bears multiple voices, that each word brings about other lives."[2] In fact, most authors of the texts contained in the anthology have multiple cultural belongings that enrich their literary production and influence their language choices, as we will see in the analysis section below. The texts are written by English-born writers based in Scotland and Asian and African-origin novelists and playwrights, including a British novelist with refugee origin, a Singapore-born author who migrated to Australia, or a Jamaican writer living in France and writing in English, among others. These multiple belongings and diverse identities have an impact on the way they represent reality through fiction, which themes they select, and what intention lies behind each text.

Similarly, multilingualism and its effects cannot be fully understood without bearing in mind the pragmatic dimension of the text and its impact on the reader. Considering the third set of questions Bar-Itzhak (2021) proposed to tackle in literary multilingual texts, of particular interest are the questions derived from the interpretation level related to the creative dimension of literary texts. Bar-Itzhak (2019, p. 4) states that: "Every multilingual work has an 'ideal reader,' who understands all the different languages and can therefore access the text in its entirety, and other readers, for whom one or more of the languages of the text remain illegible." Consequently, "the effect on the monolingual reader is different from its effect on the ideal reader", as there are different levels of interpretation and access to the content and expressive form of the literary text. Moreover, Bar-Itzhak (2019, p. 4) concludes that "the reader is reminded, in every line, of something that is beyond their reach." In the case of translated texts, readers of the target text differ from readers of the original source text, given that their language is, by principle, another one (L2). However, in multilingual or heterolingual texts, there are possibilities of rendering the source text material, which comes in L1, L3, or others, into a wider combination of languages.

## 2. *Extrañezas cosmopolitas* and Multilingualism as a Mirror of Strangeness

This section deals with multilingualism and its presence in the selected corpus of texts that are published in the edited volume *Extrañezas cosmopolitas. Antología literaria* (2022), paying close attention to how multilingualism is a mirror of the strangeness experienced by the author, by the characters, or as a result of the narrative and its settings and themes. As will be exposed, regarding the material or corpus of study, the different texts that were selected comprise an array of emotions and situations in which strangeness is the common element, and this is particularly made explicit in the way characters express their feelings of otherness. The narratives, the short theatrical play, and the two poems share a scenario in which characters face transcendental life changes or physical, mental, or emotional migratory movements that enact different reactions in the form of assimilation, adaptation, acceptance, hostility, or marginalisation. It is of high significance to underline at this point that the objects of study are texts that have in common, as main themes, the foreignness and strangeness represented in the several literary texts; the common background to all is a multilingual and multicultural reality, and they tell stories of people's lives, stories of unbelonging, or stories of dystopian experience and remembrance.

Most authors whose works have been selected for the anthology *Extrañezas cosmopolitas* manifest feelings of loss, solitude, uncertainty, and insecurity, as well as what Kaza (2017, p. 13) called "a silence between languages". Characters and literary voices are presented in incompatible worlds and conflictive realities, where they are at risk of being silenced. Therefore, translators opt for giving voice to the "third language", the language that identifies each character, his or her own language, which is either preserved in the same form or translated into a second language, the language of the target reader, with some compensation techniques so that the Spanish-speaking readership can be provided some access to the source text content while the foreignness of the source text is explained. The result is a multilingual target text, a text that is not limited to a single language or language

variety and is an expression of communities or groups with users of diverse languages, with a major presence, in some cases, of creole languages and code-switching.

In many cases, the texts collected in the anthology belong to minor literatures, "that is, the literatures written by minority cultured in the languages of the majority" (Cavagnoli 2014, p. 165). Likewise, most selected texts fall within the category of postcolonial writing, which contains stories and experiences that generate tensions affecting the use of language: "The most interesting aspect of postcolonial writing is how it marks the difference from Received Standard English. It takes over the language of the centre and replaces it with impure language", using distinctive marks of language variance (Cavagnoli 2014, p. 166). In another chapter of the same volume (Bertacco 2014), concerning the language of postcolonial texts, Rizzardi (2014, p. 184) claims that "[t]he texts of the postcolonial world are, for this reason, always linguistically contaminated, displaying a great deal of plurilingualism, bilingualism, and diglossia".

Thus, the main method of analysis of the selection of texts entails the description of the degree of multilingualism in each text, whether the characters change or evolve while deploying a second language, which may become a third language (L3) in the translation, or whether multilingualism serves as an instrument to reflect the strangeness authors experience or intend to communicate and, by extension, characters express. Secondly, while instances of multilingualism are identified, I will introduce some doubts and concerns of translators when approaching the process of translating the literary texts of the anthology.

A preliminary question is what a multilingual text is, and a first reply is that a multilingual text can be defined in simple terms as a text worded in more than one language, with varying degrees of presence of the languages. Particularly regarding literature, Meylaerts (2010) explains that "every text is a collage of many texts in several languages in an often continuous translation chain" (Meylaerts 2010, p. 228) and that "[l]iterary multilingualism may take on numerous forms according to the quantity (one single word vs. entire passages) and the type of foreignisms used (dialects, sociolects, foreign languages, etc.)." (Meylaerts 2010, p. 227). In addition, Delabastita and Grutman (2005, p. 17) point out that "the actual quantity of foreignisms in a text is rather less important than the qualitative role they play within its overall structure, i.e., their potential as functional elements." Clearly, the number of actual instances is relevant, but exploring their function and role in each text gains more significance for the purpose of this paper. Consequently, our main purpose is, in this section, to undertake the analysis of multilingualism in the anthology of texts we selected and, in the next part, to focus on how these literary texts are translated into Spanish so that we can present and discuss the priorities and restrictions that affected the translators' decisions and their dependence on various factors. Therefore, the operations that take place in each text will be described and explained below to provide some qualitative results about the effects of multilingualism.

The corpus of literary texts included in the anthology belongs to different literary traditions, and the various contributions were originally written in English or French, sharing a common theme around the notions of strangeness, otherness, and migration in a contemporary global and so-called cosmopolitan world where the figure of the stranger emerges and lives. Marotta (2010, p. 107) distinguishes between the concepts of strangeness and the stranger: "[t]he idea of strangeness has been associated with a spatial process that describes the proximity and distance between social actors. Strangeness therefore exists when those who are physically close are socially and culturally distant." while "the stranger as a social type describes individuals who are socially, culturally or racially different from the host or dominant group." With these two notions in mind, Marotta concludes that "[t]he cosmopolitan stranger has contributed to our rethinking and reassessment of cross-cultural encounters and the emergence of a cross-cultural mode of thought, but it has also underplayed the incommensurability of cultures and the "prejudices" that are inherent in the cosmopolitan's world-view." (Marotta 2010, p. 120) Therefore, we have examined the literary texts of *Extrañezas Cosmopolitas* in light of the cross-cultural encounters and modes

of thought that take place in every text, paying particular attention to the characters that feel like strangers in different environments.

Canadian Wayde Compton's *The Lost Island* is speculative fiction that explores the link between identity and place through the experiences of some characters of various origins who inhabit Vancouver. The main characters of the story are anti-colonialist activists who, within a futuristic approach, aim to share a new territory, a recently emerged island, among Afrodescendants and indigenous peoples. The language uses predominantly scientific terminology, mostly names of materials and geological components, which brings readers into a situation of distance and strangeness from the described uncommon reality. Language register and the use of specialised language create a breach between fictional reality and the reality of a contemporary readership. The narrative is a metaphorical return to the origin of life and to the consideration of indigenous peoples as true inhabitants of the land, and multilingualism appears as a natural element of the multilingual and cross-cultural reality of Canada. Therefore, the text is rich in proper nouns of periodicals and commercial products and words with some ideological burden like "golliwog" or culture-bound names of the indigenous groups of the alluded territory.

The poem *L' élue* by Fatoutama Sidibé, an author born in Mali who later moved to Brussels and back to Mali again, is a poem of remembrance of her childhood in that African country. The text is about remembering the African starry nights or the maternal pride of her mother, among other mental images she recalls. The author tells us how she felt a stranger in Mali, even though she had been born and lived there before, as if, in a certain way, she belonged there and, in another, she did not. Multilingualism comes in the form of some interspersed African expressions that are present in the French source text, and an intentional lack of punctuation contributes to creating a poetic rhythm, which serves to reinforce her reflective mood, thereby reproducing the pace of a stream of consciousness, which will bring some challenge to the Spanish translator.

Additionally, in verse is the poem *Look at Me How*, by the Scottish writer A.L. Kennedy, which was published online in September 2020[3], as a reaction to the social and political events that derived into the Black Lives Matter movement. This text addresses white target readers, inviting them to look at themselves. She points out that one cannot control having a particular colour of skin and describes the implications of this on one's own and others' lives. The main focus is not on those treated like strangers but on target white readers, who are directly appealed to by the poet to recognise themselves as part of the norm-making system and as members of a society in which they occupy a privileged position. Throughout the narrative, a case is built upon the attitude of superiority and the position of tolerance that have prevailed in previous times, and Kennedy relates these attitudes to the sense of superiority towards others, especially those who look different.

The language is at times harsh and direct and provokes strong mental images, culminating in a climactic end. When Kennedy asks readers to look at her and at themselves, she is demanding from them an assumption of their historical and social debt. In this poem, multilingualism plays an ideological role since it contains L3 terms, which are neither English (L1) nor Spanish (L2), to incorporate some symbols and references to oppressors throughout history, which require some translation decisions when the poem is translated into Spanish. The poet feels a stranger, at a distance from the events that eventually derived in the Black Lives Matter movement, and the various terms in a L3 or L4 language remind us of the sense of strangeness felt towards the previous historical and social situations of oppression and abuse.

The *Vanity Project* is a narrative written by Alecia McKenzie, a Jamaican writer established in France after living in the United States of America, Belgium, England, and Singapore. The story takes place in Jamaica, and it is about a group of women who attend a therapeutic writing workshop in a reintegration centre. Each character expresses her strangeness with regards to the other members of the therapy group and adopts an eccentric position towards the writing project. McKenzie presents every member of the group as the victim of a neopostcolonialist and neoimperialist legacy and as participants in

development programmes and institutional projects of transnational solidarity designed far away from their own reality. Jamaican cultural terms, colloquialisms, and examples of language variation lie at the essence of this text, whose choral nature is heightened by the various narrative voices and characters that are included in the text, giving rise to a rich multilingual text.

Leila Aboulela, a writer born in Egypt who then went to live in Sudan, Scotland, Jakarta, Dubai, Abu Dhabi, and Doha before returning to Aberdeen, published *The Boy from the Kebab Shop* in 2001. In her works, Aboulela dissects Western and African stereotypes of Islam, Muslims, and immigrants, and in *The Boy from the Kebab Shop*, she tells the story of Dina, a student whose father is Scottish and whose mother is Egyptian. Dina meets Kassim, from the kebab shop, whose mother is Scottish and whose father is from Morocco. Therefore, in this text, there are intertwined stories about identities, immigrants, and the future of the younger generations. Naturally, Aboulela's text abounds in Arabic texts and British names and terms, which contribute to portraying the multilingual and multicultural reality of Britain, where strangeness is reflected in the cohabitation of characters that employ their own first languages, so that sometimes this builds up a barrier for communication. Characters inhabit a "Third space" (Bhabha 1994, p. 38), which is based "on the inscription and articulation of cultural hybridity", manifested in the use of contact languages.

Another story in the anthology that presents multilingualism as one of the main features of the text is *The Dancers*, written by Suhayl Saadi, an English-born author of Pakistani origin who lives and works in Glasgow. One of his main interests is to visibilize generations of Scots of Pakistani origin who live in Glasgow, like himself, and to explore the tensions and negotiation processes that take place in a multilingual and multicultural context. *The Dancers* is about a young woman from Glasgow, Rosh, who struggles and negotiates with the contradictions of the emotional legacy received from her parents, an Irish Catholic mother and a Panjabi Muslim father, who seek refuge in mystic rituals according to their religious backgrounds. Rosh feels a stranger in her relationship with both her mother and her father and perceives that she does not fit in with any of them.

In this text, Standard English mingles with expressions and terms in Scots and Urdu, mostly at the same level. The author thinks that this naturalisation and normalisation of various languages is an instrument to reflect the coexistence of otherness in multicultural and multilingual Glasgow. Hence, language is a sign of strangeness if it is not understood, which immediately requires positioning and some action by translators: how to make the multilingual source text a multilingual target text as well. Names of places, brand names, language variations, or concepts from the Panjabi and Irish cultures all contribute to the multilingual nature of *The Dancers*. This title itself requires a preliminary decision by the translator about whether to use a gender-marked target text or a non-gender-marked expression. This decision stems from identifying who dances in the story (the father, the young people of the community, the protagonist, her friends, etc.) and from the ideological positioning and literary expertise of the translator.

*The Dependant's Tale As Told To Marina Lewycka* is another text that is contextualised in Britain and whose author is Marina Lewycka. It can also be listened to in audiovisual format (https://28for28.org/videos/the-dependants-tale, https://shortstoryproject.com/stories/the-dependants-tale-as-told-to-marina-lewycka/, accessed on 27 February 2020) as part of *The Refugee Tales* collection. Lewycka is a British novelist of Ukrainian origin. She was born in a refugee camp in Germany, and then her family moved to England. The story is narrated by an eight year old girl who is transported from her home in Bradford to an immigrant detention centre somewhere else. Therefore, the first remarkable aspect of the language is that the story is told by a child, reality is seen through her eyes, and that reality is a cruel one since many migrant families, like the protagonist's, long wait for a right to residence and for refugee status in a place they consider their home. Therefore, translation efforts in the case of this text are directed towards dealing with the idea of home, as expressed by the girl, and involving the target reader in the emotional impact of the feeling of strangeness and lack of understanding the girl manifests.

The experience of migrants is also the main theme of Chika Unigwe's story called *Cotton Candy*, a narrative about the lives of African women arriving in Europe full of expectations and about the difficulties they encounter as strangers in mostly white communities. In addition, we can read how they are doubly racialized for being coloured and for being women. The sweetness and naivety of sweet cotton candy contrast with the unexpected destiny of the main character when she moves from Nigeria to Antwerp. The multilingualism of *Cotton Candy* is clearly evident: Dutch terms are inserted in the Standard English literary text, where these are combined with some Nigerian (Igbo) and Nigerian Pidgin English. The following question arises: what to do with the multilingual nature of this literary text when the text is translated into Spanish if the language combination functions as one more fictional element?

Multilingualism also mirrors the cultural and linguistic coexistence of migrant-born groups or contemporary descendants of migrants in Australia in Simone Lazaroo's *Songs of Love and Warning*, which is part of the novel *Between Water and the Night Sky* (Lazaroo 2023). As the main characters move between Singapore and Perth, Australia, and have grown up in different cultural environments, there are also references to brand names, place names, food and beverages, and culture-bound expressions, which appear throughout the text in different languages. Therefore, terms in different languages serve to contextualise the settings and the situations the main character, Eva, goes through. Likewise, Lazaroo inserts reflections on language and identity, especially about Eva, the girl who is born from a mother who is "a bit English, a bit Scottish, a bit French"[4] and whose father speaks Malayan, Chinese, and English and is "one of the in-between people, as the British would say". Therefore, Eva learns to adapt to the others (speaking colloquial English, French, Chinese, and Malayan with neighbours) and to become multilingual. At one point, she asks her mother, "So I was multilingual when I was young?" and her mother replies, "Not really. More like a bad mimic. Or a badly tuned transmitter". This last reply highlights that Eva does not fully belong to any of those cultures and languages, but, at the same time, she is part of all of them. There was a chance, though, when they moved from Singapore to Australia, where education is provided in English and Eva adopts a more homogeneous identity.

*Extrañezas cosmopolitas. Antología literaria* (2022) contains three other texts: Tendai Huchu's distopian *The Sale* (2012), which envisages the future of African migrants from the present perspective; playwright Mojisola Adebayo's short play *The Interrogation of Sandra Bland* (2017), a multi-choral text with various voices placing emphasis and expressing emotions while relating when Sandra Bland was arrested during a traffic stop; and Patricia Cottron-Daubigné's poem *Énée de Syrie*, part of the volume *Ceux du lointain* (2017), which collects stories of migrants struggling against adversity, comparing them with classical myths.

However, even though they all deal with different forms of facing strangeness and becoming strangers, these three texts are not examined here since they are not multilingual texts, as we have defined them above, i.e., a text written in more than one language.

### 3. Multilingualism and Translation Challenges in *Extrañezas cosmopolitas. An Anthology*

Once we have briefly introduced the presence of multilingualism in the selected corpus of literary texts and before examining how these heterolingual texts have been translated into Spanish to be published in the *Extrañezas cosmopolitas* anthology, some preliminary approach to the profile of the translators of the collected volume should be considered. Concerning the profile of these translators, it is noticeable that they are all literature experts who selected both the texts and the authors that made up the collection for the volume *Extrañezas cosmopolitas*. This bears some significance since the translators already have a thorough knowledge of both the source text and the original authors, which facilitates the identification of signals of strangeness in the literary source texts. In some cases, as they are contemporary literary texts, the translators have established contact with the writers so that they can ask for clarification or comment on their decisions taken during the translation process. Moreover, coediting the volume demanded the provision of some

uniform translation strategies for all the texts and opting for a consistent approach to multilingualism, conceiving this as the presence of more than one language (Corrius and Zabalbeascoa 2011).

The study of the translation operations entails the consideration of the diverse factors that characterise each text and have an impact on the translation process and reception. As Hitzke claims, "[t]he translation's and translator's presence is thus not to be found in the process of reception alone, for the text itself also provides insights into these matters." (Hitzke 2016, p. 436). Thus, I will examine how multilingual instances in every literary text of the selection are translated into Spanish so that we can draw some qualitative analysis and some preliminary conclusions out of this study. Translation processes have to be interpreted bearing in mind that all the translated texts are concerned with manifesting the author's or characters' vital experiences of (un)belonging to a certain labelled culture or identifiable group, often from a diasporic point of view. Therefore, the role each word or expression plays within the text and what effect they may have on the target readers are relevant aspects for translation purposes, what Corrius and Zabalbeascoa call "the nature of the problem and the priorities involved" (Corrius and Zabalbeascoa 2011, p. 123). Secondly, it was necessary to make decisions about the translation strategies that may be employed to (re)produce the same or similar effect in the target text.

Following Corrius and Zabalbeascoa's (2011) categories of L1 (source language), L2 (target language), and L3 (a language that is neither L1 nor L2) and the multiple operations that may be possible in translation, some options are identified: (1) that the word or expression is left in the L1, with or without some explanation; (2) that it is translated into the L2, with or without some additional information; and (3) that it is transferred into the L3, with some compensation strategy, such as including a glossary or a footnote.

For the purpose of providing details of the analysis of the Spanish target texts that are part of the anthology *Extrañezas cosmopolitas*, taking Voellmer and Zabalbeascoa's words, "L3 analysis also involves tackling issues such as stylistic language variation, the presence of different discourses, and strategies for portraying foreignness and otherness" (Voellmer and Zabalbeascoa 2014, p. 239).

I will use different labels to refer to the use of L3, a language that is neither English nor French, the source languages of the original texts, and textual examples of what has been the decision of translators: whether to omit, compensate, or reproduce in a way that differs from the source text. Corrius proposes a classification of L3 TT "according to the degree or nature of manipulation" (Corrius and Zabalbeascoa 2011, p. 122): "(i) unchanged (L3TT is the same as L3ST), (ii) neutralized (L3ST is the same as L2, so they cannot be differentiated in the TT), and (iii) adapted (the L3TT is not L3ST nor has it been neutralised or omitted)." (Corrius and Zabalbeascoa 2011, p. 122). Therefore, the translation commission entails giving voice to the characters and their surrounding worlds in the Spanish target texts, keeping in mind that "there is no single way of translating an L3 because there are different variables that can influence translator's decisions" (Corrius and Zabalbeascoa 2011, p. 122). The final goal of the translation is to bring the strangeness in the source texts to the target audience and to deal with the multilingual presence in the original texts, i.e., preserving the foreign as foreign determines the translation process and product.

Moreover, the translators face the responsibility to produce a target text in Spanish preserving the same effect of strangeness as the source text, even though the degree of importance of this strangeness is a subjective one, and, as Delabastita and Grutman point out, the significance of preserving this effect "needs to be defined in pragmatic terms (e.g., depending on circumstances, the same translation error can have a fatal cost or simply pass unnoticed)" (Delabastita and Grutman 2005, p. 19).

Since language is an indispensable instrument to express the author's intentions, translators make decisions about how difference and otherness can be reflected through language or about whether the original intention of the literary author was to show alterity through language play. Within this context, translation is placed in an uncertain and uneasy

position, in which characters need to belong to some reality and language; they cannot stay foreign or "outside reality" (Neergard 2021, p. 8).

Compton's dystopian fictional story *The Lost Island* has been translated as *La isla perdida* and presents some L1 and L3 instances of interest for the purposes of this paper. The minute and detailed description and the specialised vocabulary, particularly that related to geology, add extra difficulty to the translation process. With this, Compton manages to recreate an environment characterised by the diversity of materials, forms, and substances that conform to the primary volcanic island that has emerged, some kind of mirror of the diversity of a multicultural indigenous territory like the Canadian one. In addition, this geological jargon contributes to enhancing the perception of strangeness towards an unknown reality that is ready to be explored and colonised. Thus, the translator has taken special care to preserve the specificity of this scientific terminology since it plays a fundamental narrative role, as in this example: "Hyaloclastite breccia. Opaque petrology. Co-ignimbrite plumes. The language sits there and stares back" is rendered as "Brecha hialoclastita. Petrología opaca. Nubes de coignimbrita. El lenguaje está ahí y devuelve la mirada." (Valdés and Prada 2022, p. 335).

Likewise, English proper names of periodicals, place names, or proper nouns of institutions are used to contextualise the story in a Canadian context, and this justifies their transfer unchanged in the Spanish target text: "The Sun", "MacLean's" or "UBC". The translator also decides to adapt and preserve in unchanged form the L1 and L3 terms of words that refer to the Canadian postcolonial reality, for instance, the names of indigenous peoples like Snohomish or Mohawk. Another element that remains in L3 in the target text are the Latin expressions of the source text in order to fulfil an equivalent referential function to a scientific register. Formal marks such as italics or capitals are also maintained in the target text for additional emphasis or extra attention from readers.

On the contrary, translating *The Lost Island* involves making decisions about words with an ideological burden, so that the translator carefully tackles the rendering of "Indian land" as "Tierra india" and decides to explain his decision in a footnote (Valdés and Prada 2022, p. 334). For "golliwog", he does not eliminate it or adapt it, but includes the term in the L3 and explains in a footnote its meaning in Spanish and the effect of using it in the text (Valdés and Prada 2022, p. 340): a pejorative expression for dark-haired foreigners that originates in the British colonial period. In this case, the presence of untranslated terms with an ideological burden reinforces the strangeness of the characters in the narrative experience and the potential freedom the new island offers them.

*L'élue* is a poem by Fatoumata Sidibé, rendered in *Extrañezas cosmopolitas* as *La elegida*, which transports readers to the memories of the poet about Mali and her feeling of strangeness in Brussels and in Mali once she returns there. The translator reproduces the same rhythmic pattern of the original French poem by translating the initial verbal form "Je me souviens" as "Recupero", and by omitting final punctuation marks, as in the source text. Sidibé recalls her African remembrance by evoking images with L3 words, and, in order to underline these memories, the translator preserves terms such as *galamas* in L3 and also in the source text with a French footnote that is translated into Spanish in the anthology. Additionally, terms such as "karité" or "peul", with traces of the African colonial past, are also left unchanged in L3, in italics, and some explanation is inserted in the target text, so that the multilingual nature of Sidibe's poem and the West-African foreign flavour are kept in the translation.

The title of A.L. Kennedy's poem *Look at Me How* has been translated as *Mírame cómo*, and this sentence corresponds to the initial line of the first verse in every page of the poem, except for the last verse in the final page, where this invitation becomes a dramatic question ("And why look at me and my whiteness?") that reverts the established racial hierarchies: it is high time to look at white people like her and assume responsibilities about the shared past. The poetic voice feels a stranger in her context and asks the others to look at her; she insists that she is white, a Scottish white. Therefore, the translator believes it is necessary to maintain this insistence on whiteness in the target text while also preserving the rhythm of the poem, which stems from the repetition of key terms such as the final line at every verse:

"Es simplemente mi piel, con la que nací [ . . . ] nunca supe que [ . . . ]". This repetition triggers a particular aesthetic effect on readers and insists on the idea that her skin makes her different, so that the translator replicates the same pragmatic effect on the receiver.

In *Look at Me How*, the author reconstructs the history of racial and colonial oppression through the insertion of lexical items that refer to that oppression of the white against the black, thereby expressing her strangeness and restlessness since she shares the colour of the skin with the oppressor. Related to this, there are also allusions to place names or episodes of such tragic history in the source text that are translated or replaced by their equivalent in Spanish (L2), with the exception of the Scottish bagpipe, the *piob mohr*, that is left in L3, as in the source text, with an explanatory note added by the translator about this Scottish traditional symbol that reinforces the terror produced by the army in the past. The "Klan" or the terms used for women at the service of others, like "Mensahibs" and "Karens" are also rendered untranslated: "Mírame cómo soy blanca como las Memsahibs y las Karens." (Valdés and Prada 2022, p. 370).

Translating Alecia McKenzie's *Vanity project* has been itself a complex project, given the different narrative layers and the multiple voices that intervene in the text. Translation becomes here an instrument to express in L2 the various voices of women in critical situations who feel strange and eccentric as victims of violence against women. The translator has to make decisions about how to convey culture-bound terms such as brand names like Red Stripe beer, a pale lager manufactured in Jamaica, whose name is rendered in the target text in its English copyrighted form as a symbol of identity,[5] while a footnote is added in the translation. Jamaican references are adapted or preserved throughout the Spanish text in local names like Azzan's Bazaar ("Bazar de Azzan") or in Jamaican Patois expressions such as "pickney" to refer to a child regardless of its racial origin or "kissing her teeth", an expression of disapproval by making a sound sucking air through the teeth. This language variation comes in translated form in L2, in Spanish, so the cultural bond to Jamaica, the setting of the narrative, is lost: "pickney" is rendered as "hija" and "kissing her teeth", as "chascó los dientes". Adaptation is also employed with the aim of triggering the same effect as the English source text, so that the references to violence are faithfully translated to reproduce the same emotionally cruel impact of the various terms: 'apuñalamiento', 'paliza' or 'descuartizamiento' (Valdés and Prada 2022, p. 280) for 'knifing, battering, chopping up".

Colloquialisms and onomatopoeic sounds also contribute to underlining the different emotions McKenzie aimed to create and, by extension, the orality of Jamaican language and traditions. Therefore, the translator carefully selects the Spanish equivalents to preserve the same effect: "¡Yupi!" for "Whoopee", "¡PAF!" for "BAPS!" or "claclaclá" for "clank-clank-clank". It is worth mentioning the example of the source text "dutty whore", a case of coarse language that transcribes a word pronounced with a West-Indian accent ("dutty" for "dirty"). The harsh effect of the insult has been maintained in the functional equivalent "puta sucia" (Valdés and Prada 2022, p. 292); however, there is no evidence that the language variation is explained, probably considering that the linguistic variation is secondary to the violence the author wished to remark. Regarding colloquialisms, *Vanity Project* is rich in colloquial oral speech, which comes together with the dialogues and narrative voices that can be heard in the text ("Baby"/"Nena", "Oh, yeah?"/"¡Ah! ¿Sí?" or "her thing"/"lo suyo"). These oral expressions are translated in L2, deleting any L1 or L3, but they perpetuate the oral traditions and communication features of the Jamaican culture and language tradition.

A final note should be made on a phonetic play on words that is meaningful in the story since one of the characters' mother uses it to mock her daughter's expectations of marriage when she meets a North American partner. The mother reacts by pronouncing the following phrase with the originally italicised words to identify the play on words: "Then she left home when she was eighteen, moved in with a man twice her age and afterwards moved to *Minneapolis, Minnesota*. Of all places. *Many apples, many sodas*, my mother used to laugh." When "Many apples, many sodas" is pronounced loud, this provokes a humorous

effect in English as it imitates the North American pronunciation of the phrase. However, this effect is difficult to recreate in Spanish. Thus, the translator leaves the English words untranslated and compensates the target reader with an explanation in a footnote about this ironic paronym that stems from the oral form of the English words: "Luego ella se fue de casa cuando tenía dieciocho años, se fue a vivir con un hombre que le doblaba la edad y después se mudó a Minneapolis, Minnesota. Mira que no hay sitios. Many apples, many sodas, se reía mi madre." As can be seen, the L1 terms are retained without any visual mark to distinguish them from the rest of the lexical items, and the following footnote provides an additional explanation for it: "Adaptación irónica parónima de los términos topográficos originales (/ˈmeni ˈæpəlz, ˈmeni ˈsoʊdəz/) En español", «Muchas manzanas, muchas sodas». This pun merits a detailed explanation while it produces a multilingual example in the translated text since it is a textual chunk in L1 with additional information in L2, which prioritises the target language. Semantically, this pun actually plays some fictional role, as it reinforces the few options faced by these women, of low income and education, without many expectations, and the reaction of the other characters, encircling them in a community of strangers.

In *The Dancers*, we have mentioned above that the first decision for the translator was whether to use a feminine plural form ("las que bailan") or a masculine plural form ("los que bailan") in the Spanish title. The final choice in Spanish, *Quienes bailan*, is inclusive of all the different characters that actually dance in the story, since dancing becomes an escape mechanism and an expression mode at different levels: from the main character's father's raving to the dancers in the disco bar. In this text, Saadi shows the contradictions of an emotional and culturally diverse legacy that derive from the frequent use of terms from the Panjabi heritage of the main character's father and from the Catholic tradition her mother belongs to. In this case, the author introduces words in Urdu or Panjabi (L3) without marking them with italics, so that they remain on the same level, without any distinction, as the English text (L1). In this manner, the author stresses the capacity for integrating various language(s) in cosmopolitan and multicultural contexts such as Glasgow, the setting for *The Dancers*. When asked by the translator, Saadi agreed to include a final glossary to make terms and concepts familiar to the Spanish reader, but he manifested his opposition to the use of footnotes as they may interrupt the reading process and expressed his preference for a final glossary.

Moreover, the translator has successfully transplanted to the Spanish text the musical ambient, the activity in nightclubs, fashion, or hair style (ex., the mop-top is preserved unchanged in the translation), which are characteristic of Glasgow as a cosmopolitan place where people may feel strange with respect to others. Language can be a sign of strangeness if it is not understood, and this poses an evident challenge to translators: how to make a multilingual target text from a multilingual source text without introducing any visual indicator (i.e., italics) or additional explanation on the same page. For instance, in *The Dancers*, we find terms such as "gora" and "goree", Urdu words to refer to a light-skinned man and woman, respectively. The dancing club, formerly a church, now holds a celebration of modernity and is compared with a mystical place: "Khoon aur mitti. Faisalabad. La Casa del Juicio". The translator respects the multilingual nature of the source text and preserves the expressions "Khoon aur mitti", which she explains in the final glossary, and "Faisalabad", which is a known city in Pakistan. In contrast, she adapts and translates "House of Judgement", which is an L1 item, into L2 "La Casa del Juicio".

Regarding the family and interpersonal relations between subjects belonging to different social, religious, and cultural groups, the translator transmits a feeling of in-betweenness, a hybrid sense of identity, by making linguistic and cultural adaptations in episodes such as when the mother is praying with a rosary or the father dances like a ceviche. Numerous words in Urdu or Panjabi are shown in the text ("bhai", "kisaan", "djiin", "ustaad" or "mutkae ha pendtha", among others). This multilingual approach can be interpreted as a narrative technique to convey the daily lives of the story's characters, who are exposed to a feeling of cultural and language strangeness. The best exponent

in the narrative of this sense of strangeness and in-betweenness is the character of Rosh, whose dual identity is particularly reflected in her having two names: "Cuando se sentía irlandesa, era Róisín Dhu, la rosa negra, y cuando pensaba que era de Faisalabad, se convertía en Roshani, el rayo de luz. La división la atravesaba por dentro, la fraccionaba en dos, distribuía sus lealtades." (Valdés and Prada 2022, p. 56) Consequently, Rosh admits having difficulties understanding symbols of identity such as the green colour in the Glasgow Rangers matches: is green a symbol of the hara of Punjab, or does it symbolise the Republican banner of West Belfast? How do I reconcile both positions? Multilingualism reveals that languages and cultural elements can live together in the same text, just as they can coexist in one person.

However, when translating *The Dancers*, the analysis showed that, although words in Urdu and Panjabi were left unchanged and unmarked, those in Scots, which implied grammatical and syntactic marking, were lost in the Spanish translation. Related to this, Ashley (2011) precisely pointed out what we can call a paradox:

> Urdu greetings, which do belong to a majority language (there are 400 million Urdu speakers), remain "other," untranslated, and italicised in French, while the Scots narration is normalised. What this reveals is that it is easier to demonstrate that Urdu is "other" in a Scottish context than it is to translate the fact that Scots narration is "other" in the context of a so-called English-language novel by a Scot. In other words, the most distinctly local aspect of Saadi's text, the play of immigrant languages against a dominant Scottish voice, is lost when it is transferred to another cultural and linguistic context.

The coexistence of Standard English with Scottish English and colloquialisms from the area of Glasgow is not rendered in the Spanish translation, so language variation is largely dissolved in the target text, and thus the target audience cannot perceive it.

In our Spanish translation *Quienes bailan*, we also lose the "dominant Scottish voice", but it gains in the preservation of otherness and strangeness by maintaining most Urdu and Panjabi expressions without any intervention of the translator until the end. Consequently, the translator has been actively involved in bringing into Spanish the author's intended mission and the role assigned to multilingualism. In addition, unmarked terms in the L3 of a text do not require the active participation of the reader to move forward to an endnote or footnote, and thus the reader is led to complete the understanding of the text. Marking differences or the intrusion of footnotes would have interrupted the aesthetic and creative appreciation of the literary text.

Chika Unigwe published *Cotton Candy* in English in 2009, about a Nigerian young woman who arrives in Antwerp and, cheated by the migration mafia, ends up as a prostitute. The author tells us, readers, that we cannot trust the narrator as she may be lying to us and to others, which is, in a certain way, a mode of empowering her and giving her the power to manipulate the story. The English source text is interspersed with words in Dutch, English, Nigerian Igbo, Standard English, and Nigerian Pidgin English, mirroring the multilingual and multicultural context she inhabits. Therefore, references are made to cultural elements of Nigeria, for instance food and dishes, and to certain expressions in Dutch such as "een jonge afrikaanse" or "Hallo, mijn schat" to emphasise and establish bounds between the protagonist's Dutch experience and prostitution. Concerning the translation, the translator has decided to preserve the L3 of the source text as L3 in the target text, given its role in the narrative. Only in a few cases has she opted for amplifying the meaning of certain expressions in the final glossary when their meaning cannot be presupposed from the L3 term: "Mmonwu Christmas", "Enugu", "akara", "atilogwu", "¡akpuobi!", "tomapep", "Afrikanen" or "Nog een prettige weekend, mevrouw". The final explanations inserted in the Spanish source text intend to position the reader in a multilingual situation, to receive the same impact as the reader of the source text, and to generate cultural shock and uneasiness.

In *The Dependant's Tale As Told To Marina Lewycka*, the story recreates the experience of a girl while she was being transferred to several immigration detention centres from the

place where she had lived, so that she expresses her feelings of strangeness when she is far from her home. Thus, the translation process of this text focuses on the way to transmit these feelings with the language of a girl, who uses a very explicit and direct style by repeating words like "stranger" against "home" in Lewycka's English text.

One of the main translation challenges is to render the violent effect and atmosphere when the officers ("men in uniform") irrupt in their home, maintaining the tension and rhythm of the narrative of the source text. Words such as "miedo", "pánico", "rabieta", "gritar", "llorar", "chillar" or "dar golpes y gritos" (Valdés and Prada 2022, pp. 180–81) provide a very precise impression of what a young girl thinks, and thus the translator has managed to preserve the direct and simple style. Even though there is only one example of an English acronym, LSE, that has been retained and amplified in the Spanish text as L3 ("London School of Economics"), probably because it is a familiar term for the Spanish readership, multilingualism is not a significant issue in Lewycka's text.

*Songs of Love and Warning*, by Simone Lazaroo, concerns the notion of in-betweenness, which affects Eva, a girl, her relatives, and her neighbours, the main characters of the story, which takes place in Singapore and Australia. The translator has identified the linguistic and cultural references that bring the Spanish reader towards the multilingual and multicultural reality of the characters, particularly in those episodes when the colour of the skin is a sign of otherness and difference, or in the detailed descriptions of the family's origins, or in the social and geographical background of the characters.

Unlike the previous story, the text abounds in culture-related terms that remain foreign or distant for the Spanish reader, and this makes the translator opt for reproducing them in italics, as in the case of proper names of Australian periodicals and advertising slogans, without any additional explanation. On the contrary, food terms are transferred unchanged to the target text, with corresponding entries in a final glossary. This translation strategy seems to be adequate to make Spanish target readers perceive the multilingual and multicultural nature of the text and of the group of characters that are influenced by the sense of strangeness of the Australian territory, once they live there. Additionally, the Spanish reader benefits from the final glossary to clarify and obtain a thorough comprehension of the text, although in most cases, the context provides the necessary cues to understand the content. In the case of the L3 expression "lah", the translator explains the meaning and function of the term,[6] highlighting that it is an evident indicator of the Malay identity of the father, who tends to use this expression quite often. Leaving this colloquial expression untranslated places emphasis on the position of strangeness and the social isolation and contempt this character experiences, reinforcing his sense of difference, which lies in the colour of his skin ("the yellow peril").

Thus, multilingualism is an essential instrument of featuring Australia as a multicultural territory, with all the tensions and advantages this may have, so that abundant references to brand names, place names, or food terms are either left in English (L1) or in French, Malay, and Chinese (L3, L4, and L5): "kuey teow" (Valdés and Prada 2022, p. 201), "jalebi" (ibid., p. 201), "papadam" (ibid., p. 201), "roti canai" (ibid., p. 201), "nasi lemak" (ibid., p. 201), "kampong" (ibid., p. 211), or "Hantu Maligang" (ibid., p. 212), whose meaning is explained in a final glossary.

Aboulela's *The Boy from the Kebab Shop* also contains many cultural and religious references for which the translator must find some adequate rendering in Spanish, whether to preserve the L1 term or to use a common equivalent expression shared by the target reader. Examples of brand names of commercial products like Pampers or Panadol have been replaced with cultural equivalents in the target culture (Dodotis and paracetamol), while television programmes such as Weight Watchers or Al-Nihaya have been rendered in English and Urdu. For pagan and religious festivities like Hogmanay, equivalent to the Spanish Nochevieja, the translator has maintained the Scots word and amplified the text by adding an explanatory note. Therefore, the presence of the L3 terms in the original texts serves to reinforce the identitarian duality of the characters of the story, who experience strangeness between two cultures: the Pakistani and the Scottish ones. Consequently,

references to traditions and social usage of the Pakistani community or to Islam have been rendered as they appear in the source text, with an additional note at the end of the story: "khawagah", "Insha Allah" and "Salaamu Alleikum". In some cases, what used to be L3 expressions like curry or kebab are rendered in the target text since they have already been imported to the Spanish-speaking cultures and are accepted with the Spanish graphic form of curri or kebab, according to the Spanish Language Dictionary (https://dle.rae.es/, accessed on 23 May 2023).

## 4. Conclusions

The previous analysis of these representative literary texts on strangeness from *Extrañezas cosmopolitas. Antología literaria* (2022) provides an overview of some of the main translation strategies in multilingual contemporary literary texts, where the presence of more than one language is a reflection of the inherent diversity and foreignness of either characters, narrators, or stories. Moreover, this multilingualism becomes the priority that determines one mode of translation or another, and it can clearly be identified as a decision-making factor.

As has been explored above, in the Spanish target texts, culture-bound elements such as titles, brand names, place names, or food names tend to appear in L3, a language that is not the predominant language of the given literary text, which in this anthology is either French or English. Thus, in the Spanish translation, this kind of term tends to be preserved unchanged with some additional marking in the form of italicised writing, together with an explanatory note of its meaning (e.g., italicised "khawagah").

An adaptation or transcreation of the word is also a common translation strategy, where the source text word undergoes a grammatical adaptation to the L2, or predominant language, of the target text (e.g., "cups of Bushell's tea" translated as "tazas de té Bushell" and "Gezira Club" rendered as "el Club Gezira"). In these cases, a translator's note or inclusion of the term in a glossary is not necessary since the reader can interpret the meaning of the adapted form while also recognising the otherness and strangeness of the original context these words belong to.

Another translation strategy consists of the insertion of a glossary at the end of a literary text, which is arguably one of the most interesting options to render a multilingual text in such a way that the combination of languages is not perceived as a switch from one code to another (code-switching) during the reading process but as a naturalised flow of textual material and the coexistence of various languages. One of the authors in the *Extrañezas cosmopolitas* anthology (Valdés and Prada 2022), Suhayl Saadi, concerning this anthology, explicitly stated to the translator that all words should appear graphically equal in the text, and his wish was obviously respected in this edition. Regarding this issue, in an interview in 2009, Saadi provided the following explanation about his position:

> I also deploy the rhythms and ambiences of other languages in the text. Urdu, Arabic, and Persian employ multiple narrative and symbolic levels—this is one of the reasons why poetry written in these tongues is so difficult to translate effectively—and I try to employ some of that in my writing in English. I am then, attempting, to shift the gravitational possibilities of the English language, again, in terms of potential thought processes.

> Words—sometimes non-English words—are sometimes inserted to underpin, yet simultaneously disrupt and re-direct, the flow.

> (Erskine 2009)

From the study of the translation of the corpus, the idea that there is no single way of translating an L3 because there are many variables that can influence the translator's decisions has been reinforced. As Corrius and Zabalbeascoa (2011) claim, "it should be noted that there is no single way of translating an L3 because there are many variables that can influence translators' decisions. Several solutions are plausible depending on the aims of the translation and the translators' priorities and restrictions (textual, stylistic,

ideological, linguistic, cultural, among others)" (Corrius and Zabalbeascoa 2011, p. 122). As has been proven throughout the analysis, these variables help translators choose among possible solutions, such as leaving the L3 text unchanged, neutralising it, or using an equivalent or adapted form.

The examination of the collected source texts and their translations also reveals that multilingually translated texts that are translated from a multilingual and multicultural perspective can make visible the otherness, the strangeness, the multiple voices, and the realities that exist within literary works. This is so because translation places emphasis on diversity and uniformity at the same time, blurring linguistic and cultural differences in the multilingual settings and contexts that are represented in literature. In this sense, Vidal Claramonte (2021a) claims that she understands translation not as the transmission of contents but as a transnational territory that establishes a relation with the foreign, and she proposes "a cosmopolitan translation that highlights the multiplicity of languages, the cultural other, and is therefore against ethnocentrism, racism, cultural narcissism and imperialism." Undoubtedly, there is an ethical side to the translator's positioning to respect the otherness in multilingual texts, which is worth exploring in future studies.

This multilingual approach to translation changes the theoretical vision of the nature and function of translation and contributes to giving voice to more languages and mirroring the strangeness that can be found reflected in texts, literary and non-literary. As Bielsa (2016, 2023) points out, the contribution of cosmopolitan strangers promotes a new approach to translation, understanding it "not as a transfer or communication of meaning, but to a much wider notion of translation as a social relation across linguistic difference." (Bielsa 2023, p. 136) This author concludes that translation is, in fact, "a way of mediating distance." (Bielsa 2023, p. 149). In this study, translation has been approached from a point of view that is aligned with Antoine Berman's consideration of translation, in Rizzardi's terms, as "accepting the Foreigner as a Foreigner, accepting the Other precisely as Other" (Rizzardi 2014, p. 181). Thus, multilingualism facilitates the presence of the inscription and the perception of the Other in translated texts when this is part of the mission of the translation.

Contemporary fiction and fictional translation still have multiple possibilities to explore on how to render foreignness, otherness, and strangeness through multilingual texts, not only literary ones but in multimedia format. In the case of literature, multilingualism and translation bring to light hybrid identities instead of divisions and diversity, as well as provide an insight into the global challenges affecting the production and circulation of literary texts. As Damrosch stated regarding world literature, this is 'writing that gains in translation' (Lenfield 2019, p. 281).

**Funding:** This research was funded by the Spanish Ministry of Science, Innovation and Universities. Project "Strangers and Cosmopolitans: Alternative Worlds in Contemporary Literatures." Reference: RTI2018-97186-B-100.

**Informed Consent Statement:** Informed consent was obtained from all subjects involved in the study.

**Data Availability Statement:** Data for this study received the approval of the authors of the texts and of the editors of the volume that contains the translations. They are available in published form.

**Conflicts of Interest:** The authors declare no conflict of interest.

## Notes

[1] "Strangers and Cosmopolitans: Alternative Worlds in Contemporary Literatures." Funded by the Spanish Ministry of Science, Innovation and Universities. Reference: RTI2018-97186-B-100.

[2] "[C]readores de cualquier parte del mundo que no se expresan necesariamente en la lengua del lugar que les vio nacer; escritores de pertenencias múltiples, de identidades líquidas, de archipiélagos criollizados. Escritores desterritorializados, nómadas, que nos recuerdan que el lenguaje es muchos lenguajes, que cada uno de nosotros somos portadores de múltiples voces, que cada palabra trae consigo huellas de otras vidas." (Vidal Claramonte 2021b, p. 14).

[3] http://thecommonbreath.com/lookatmehow.html (accessed on 23 May 2023).

[4] We cannot provide page numbers for the quotes, since the author granted us with permission to use an unpublished version of the text.

5   https://www.redstripebeer.com/agegate?redirect=/ (accessed on 23 May 2023).
6   *Lah* "hace referencia a un coloquialismo ampliamente empleado entre la población de Malasia" and "se utiliza para indicar la actitud del hablante en diferentes contextos (sorpresa, exasperación, exclamación o rechazo, entre otras)".

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
