# Peer review of "Multilingualism as a Mirror of Strangeness in the Translation of Contemporary Literary Texts"

_languages, doi:10.3390/languages8020140_

Round 1

Reviewer 1 Report

This article is well-written and well-structured. The objectives are clearly stated, and all the information presented in the theoretical framework is well organized and clearly stated. The references used are relevant, and many of them are up to date. For all these reasons, I recommend the publication of the paper. I would only advise the author/authors to include the following reference that I believe may be relevant to their study:

Moreno Tovar, M. (2020). (A)bridging the Gap – A study of the norms and laws in the intralingual translation of the novel And Then There Were None by Agatha Christie. Revista De Lenguas Para Fines Específicos26(1), 51-68. https://ojsspdc.ulpgc.es/ojs/index.php/LFE/article/view/1234 

Reviewer 2 Report

I would like to congratulate the authors for this article and add some minimum suggestions that may improve their text.

- On the Otherness and its relation with translation, I would highly reccomend some references to "Derrida, the Subject and the Other: Surviving, Translating, and the Impossible", by Lisa Foran, where the question of the strangeness, the Other and the multiplicity of voices is thoroughly analysed. On the same page, "Translating Borrowed Tongues. The Verbal Quest of Ilan Stavans", by Vidal Claramonte is also a very recent reference they could add when reflecting on translingualism.

- The text could also benefit from a transversal reflection on the ethics of respecting the otherness in multilingual texts.

- If the author wants to improve the clarity of the analysis presentation, they might add some tables to present original-translation solutions instead of introducing all the elements in the body of the text, which at some points make it difficult to follow the overall translation strategy.
